# Emulsion Templated Porous Poly(thiol-enes): Influence of Photopolymerisation, Emulsion Composition, and Phase Behaviour on the Porous Structure and Morphology

**DOI:** 10.3390/polym14071338

**Published:** 2022-03-25

**Authors:** Viola Hobiger, Muzafera Paljevac, Peter Krajnc

**Affiliations:** PolyOrgLab, Faculty of Chemistry and Chemical Engineering, University of Maribor, Smetanova 17, 2000 Maribor, Slovenia; viola.hobiger@um.si (V.H.); muzafera.paljevac@um.si (M.P.)

**Keywords:** high internal phase emulsions, thiol-ene polymerisation, polyHIPE, bicontinuous structure, phase inversion, poly(thiol-enes)

## Abstract

1,6-hexanediol diacrylate (HDDA) or divinyl adipate (DVA) and pentaerythritol tetrakis(3-mercaptopropionate) (TT) were polymerised via a thiol-ene radical initiated photopolymerisation using emulsions with a high volume fraction of internal droplet phase and monomers in the continuous phase as precursors. The porous structure derived from the high internal phase emulsions (HIPEs) followed the precursor emulsion setup resulting in an open porous cellularly structured polymer. Changing the emulsion composition and polymerisation conditions influenced the resulting morphological structure significantly. The investigated factors influencing the polymer monolith morphology were the emulsion phase ratio and surfactant concentration, leading to either interconnected cellular type morphology, bicontinuous porous morphology or a hollow sphere inverted structure of the polymerised monoliths. The samples with interconnected cellular morphology had pore diameters between 4 µm and 10 µm with approx. 1 µm sized interconnecting channels while samples with bicontinuous morphology featured approx. 5 µm wide pores between the polymer domains. The appropriate choice of emulsion composition enabled the preparation of highly porous poly(thiol-enes) with either polyHIPE or bicontinuous morphology. The porosities of the prepared samples followed the emulsion droplet phase share and could reach up to 88%.

## 1. Introduction

Emulsion templating is a well-established method for obtaining macroporous polymers with an interconnected cellular porous structure where droplets of the emulsion’s internal phase do not contain monomers but form the template for the macropores in the yielding monolithic polymer material. The interconnecting pore structure is formed due to the shrinkage of polymer film during the gelation [1,2,3,4]. For the generation of interconnected cellular porous structure emulsions with monomers in the continuous phase and a high volume fraction of the droplet phase (typically over 70 vol%), high internal phase emulsions (HIPEs) are used, and the resulting polymers are termed polyHIPEs (see Figure 1). Since the precursor emulsion dictates the morphology of the resulting porous monolithic polymer material, the stability of the emulsion represents the most important factor for the polyHIPE preparation. The most commonly utilised methods of stabilisation include surfactants, usually non-ionic ones and particles. The porous, cellular topology of polyHIPEs is advantageous for many applications of the materials, e.g., biomedical applications [5,6], as catalytic support [7,8], or separation and purification [9].

Most polyHIPEs so far have been prepared using a thermally induced chain-growth polymerisation process, while step-growth polymerisations are less common. Recently, thiol-ene photopolymerisation was successfully introduced into the emulsion templating technique [10]. The method has proven to be particularly useful due to rapid curing times and high yield. Furthermore, samples are typically less brittle than the common styrene-based polyHIPEs. Due to ester linkages introduced via the thiol molecules, thiol-ene polyHIPEs are susceptible to hydrolysis and thus appealing for biomedical applications. Examples include the preparation of polyHIPEs from radical thiol-ene photopolymerisation for tissue engineering purposes [11], additive manufacturing processes [12,13], or drug delivery [14].

An emulsion can undergo a phase inversion process, apart from other destabilisation processes leading to phase separation (emulsion breakup; Figure 2). The phase inversion process proceeds through a meta-stable bicontinuous-like state. In the case of a microemulsion, a stable bicontinuous state depends on the surfactant to the aqueous phase to oil phase volume ratio and on the surfactant structure. The bicontinuous state can be used for templating the porosity in polymers yielding porous polymers with a bicontinuous topological structure. Numerous examples of the preparation of polymers with bicontinuous morphology in this way are known; however, all in the domain of meso and micropores due to the size dimensions of microemulsions [15,16].

On the other hand, in the case of a macro emulsion, the process of phase inversion (continuous phase inverts to droplet phase and the droplet to continuous) can proceed as catastrophic or transitional. Due to the droplet size, there is no thermodynamic stability and no stable or meta-stable bicontinuous state. In the case of a transitional phase inversion process, the surfactant nature (thus, its hydrophilic–lipophilic balance—HLB) changes, and it tends to stabilize oil in water rather than water in oil emulsions, thus triggering the phase inversion. In a catastrophic phase-inversion process, the volume ratio of oil to aqueous phase change triggers the phase inversion while the HLB value of the surfactant is not changed.

While the polymerisation of a monomer containing microemulsion can lead to microporous bicontinuous morphology, we were interested in obtaining monolithic porous polymers with a bicontinuous topology on the macroporous scale. Such monolithic polymeric materials could potentially be used in applications where flow through a monolithic piece, be it a column or a membrane, is required, and the backpressure should be minimised. Macroporous polymers with bicontinuous topology can also be prepared from a homogenous monomer/porogen solution via spinodal decomposition at the phase separation process [17]. Furthermore, the conditions of the photopolymerisation process can determine the morphology of porous polymers during the phase separation process, leading to either a cauliflower-like agglomerated bead structure or a bicontinuous-like one [18].

We were interested in whether a phase inversion process within a high internal phase emulsion could yield macroporous polymeric monoliths with bicontinuous topology. While a bicontinuous porous morphology is possible to achieve via phase separation of a monomer/porogen homogenous solution, the pore volumes, in those cases, are typically under 60 vol%, and the mechanical properties of monoliths prepared in this way with a higher share of solvent become too poor. On the other hand, it has been demonstrated that in the case of emulsion templating, porosities up to 80 vol% still yield monoliths with decent mechanical properties [19,20].

## 2. Materials and Methods

### 2.1. Materials

1,6-hexanediol diacrylate (HDDA, 80% technical grade, Sigma Aldrich, St. Louis, MO, USA) was passed through aluminium oxide (Al_2_O_3_, Acros Organics, Waltham, MA, USA) to remove inhibitors. Divinyl adipate (DVA, TCI Chemicals, Tokyo, Japan), pentaerythritol tetrakis(3-mercaptopropionate) (TT, Sigma Aldrich), poly(ethylene glycol)-block-poly(propylene glycol)-block-poly(ethylene glycol) (PEL-121, HLB = 1.0–7.0, Sigma Aldrich), Irgacure 819 (I819, IGM Resins, Waalwijk, Netherlands) and calcium chloride hexahydrate (CaCl_2_∙6H_2_O, Sigma Aldrich) were all used as received.

### 2.2. Preparation Procedure of TT:1,6-HDDA Polymer Monoliths

Measured quantities of 2.070 g of pentaerythritol tetrakis(3-mercaptopropionate), 2.415 g of 1,6-hexanediol diacrylate (ratio 1:1 of thiol to acrylate groups), Irgacure 819 (2 wt% to monomers) and surfactant PEL-121 (15 or 20 wt% to monomers) were added to a two-necked brown round bottom flask which was fitted with an overhead stirrer equipped with a D-shaped blade. The separately prepared aqueous phase (between 70 and 80 vol% of the total emulsion consisting of 1.76 g of CaCl_2_∙6H_2_O in 100 mL water) was added dropwise to the organic phase while stirred at 300 rpm. The stirring was continued for 30 min after the addition of the aqueous phase. Composition data for all prepared emulsions can be found in Table 1. Emulsions were transferred to a silicon mould and polymerised in a UV chamber (Intelliray 600, Uvitron, West Springfield, MA, USA, distance from light source: 130 mm, intensity at curing position: 120 mW/cm^2^) at 100% intensity for 120 s. The polymers were purified via Soxhlet extraction with 2-propanol for 24 h and then dried under air and in vacuo. Samples are annotated as M_A_B, where A is the volume percentage of aqueous phase and B is the percentage of surfactant relative to the weight of monomers.

### 2.3. Characterisation

Scanning electron microscopy images (SEM, Phillips XL-30, Amsterdam, Netherlands) were recorded at 20 kV. Samples were coated with platinum using ion sputter Jeol JCF-1100E (Tokyo, Japan) for 4 min at 70 mA. The skeletal density (*ρs*) was measured using a Micromeritics helium pycnometer AccuPyc II 1340 (Micromeritics, Norcross, GA, USA). The determined skeletal density was used when measuring envelope density (*ρe*) with a Micromeritics GeoPyc 1360 (Micromeritics, Norcross, GA, USA) to determine the porosity of the samples. The measurements were performed with a 12.7 mm cell and piston system. The porosity of the samples *P* was calculated using:$$P\;\left(\%\right)=\left(1-\frac{\rho_{e}}{\rho_{s}}\right)\times 100$$

## 3. Results and Discussion

The introduction of monomers into the continuous phase of an emulsion enables the preparation of porous polymer monoliths with primary pores templated by the droplet phase of the emulsion. High internal phase emulsions are of particular interest because, due to the high volume fraction of the droplet phase, the resulting polymer monoliths exhibit high pore volumes and interconnected open porous cellular morphology. When sufficient emulsion stability is achieved, the morphology of a polyHIPE material replicates the HIPE features at the point of gelation [21].

As we were interested in applying high internal phase emulsion templating for producing macroporous polymers with different morphologies, experiments were designed in order to study the emulsion phase inversion phenomenon, and especially the possibility of combining the photopolymerisation of a thiol/acrylate monomer mixture with the phase inversion to obtain poly(thiol-enes) with a distinct bicontinuous porous morphology.

A well-established high internal phase emulsion setup, consisting of an aqueous droplet phase, and a thiol-ene polymerisation mixture with 1,6 HDDA and TT as monomers was used as the continuous phase.

In our experimental design, we decided to focus on the surfactant and phase volume ratio, as this is the most important factor governing the phase inversion process at high internal phase emulsions [22].

The influence of the phase volume ratio was first studied. The monomer ratio was fixed at 1:1 (functional group ratio), and the concentration of surfactant to 20%. At 80 vol% of the aqueous phase (droplet phase not containing the monomers), a polyHIPE morphology (cellular interconnected) was formed (M_80_20), suggesting enough kinetical stability for the polymer film to form within the frame of the precursor emulsion. Decreasing the volume fraction of the droplet phase to 75 vol% resulted in a monolith with a bicontinuous morphology (Figure 3, sample M_75_20, Figure 4). In Figure 4, the features of this type of morphology are clearly visible, with continuous polymer domains and pores and the distinction from a cauliflower-like structure typically derived from phase separation of the polymer from solution is evident. A further decrease of the droplet phase volume to 70 vol% yielded monoliths with fused hollow bead morphology (M_70_20), suggesting phase inversion occurred prior to gelation. No significant shrinkage of the monoliths (except in the case of M_70_20) was observed. Therefore, it is proposed that the mechanism of the bicontinuous morphology creation is the emulsion phase inversion with the photopolymerisation inducing the gelation within the meta-stable state. With the lowest droplet phase volume (sample M_70_20), an inverted structure was observed.

Furthermore, the concentration of surfactant was lowered in order to study the role of surfactant concentration in the process. At 15 wt% of surfactant, the same trend was observed; a polyHIPE morphology was formed at 80 vol% droplet phase (sample M_80_15, Figure 5) while lowering the volume share of the droplet phase to 75 vol% and 70 vol% caused the shift of the morphology of the resulting monoliths towards the bicontinuous and hollow sphere, respectively (Figure 5).

While changing the surfactant concentration from 20 to 15% did not alter the trend of morphology change from polyHIPE to bicontinuous and inverted with decreasing the volume share of the droplet phase, we were further interested in the influence of a more significant surfactant amount alteration at a fixed droplet volume share of 80 vol%. Samples were prepared, with surfactant amounts from 5% (M_80_5), 15% (M_80_15), and 30% (M_80_30) (Table 1, Figure 6). Increasing the amount of surfactant at 80 vol% of droplet phase (a typical high internal phase emulsion) causes the change of morphology from cellular (15% or less surfactant) to bicontinuous (higher amounts of surfactant), suggesting that larger amounts of surfactant facilitate the formation of a meta-stable emulsion state which is reflected in appropriate gelation and the resulting bicontinuous morphology.

To study the effect of surfactant and surfactant composition on the morphological structure of thiol-ene photopolymerised HIPEs, a different monomer composition was tested as well. The established system based on the vinyl ester, DVA and TT was first reported with two surfactants present in the HIPE formulation [23]. One way of achieving a macroporous bicontinuous structure would be to vary the ratio of surfactants or leave out one surfactant altogether in order to change the HLB and overall composition of the surfactant layer. It was chosen to lower the amount of Span 65 consecutively and finally omit it to have only PEL-121 present in the formulation. At first, the original setting with 15% surfactant was reproduced, which yielded a typical polyHIPE morphology, according to SEM (see Figure 7). The SEM imaging revealed that the presence of Span 65 seems to be essential for establishing a cellular interconnected polyHIPE architecture (see Figure 8). When only PEL-121 was present, it was not possible to obtain a stable emulsion above 75 vol% of dispersed phase. However, with 75 vol% of internal phase, a monolith with bicontinuous structure was the result.

The comparison of the two systems, HDDA/TT and DVA/TT, highlights the importance of surfactant choice and composition in a HIPE. It also showed that, upon careful tuning, it is possible to obtain a macroporous bicontinuous pore architecture from various thiol-ene HIPE formulations, opening a wider range of materials and material properties for potential applications. The chemical compositions of the emulsions prepared with DVA can be found in Table 2.

We have shown that thiol/acrylate-based and thiol/vinyl ester-based polymers synthesised via a thiol-ene radical-induced photochemical process within high internal phase emulsions can exhibit different porous morphologies, from a typical polyHIPE-type cellular interconnected one, resulting from a kinetically stable high internal phase emulsion, through a bicontinuous with polymer/pore domains, to an inverted, hollow fused sphere-like morphology. This is the first example of the preparation of a polymer monolith with bicontinuous morphology resulting from high internal phase emulsions. With a careful emulsion composition setup, a thiol-ene photopolymerisation process can yield polymer monoliths with a bicontinuous morphology, high pore volume and homogeneous structure throughout the bulk of the monolith. We believe this could be beneficial in various fields of application while further investigations regarding the influencing factors and mechanism of morphological alterations are ongoing.

## Figures and Tables

**Figure 1 polymers-14-01338-f001:**
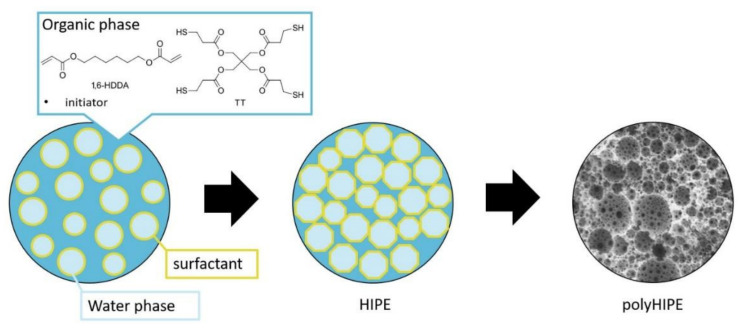
General scheme of porosity templating via high internal phase emulsion.

**Figure 2 polymers-14-01338-f002:**
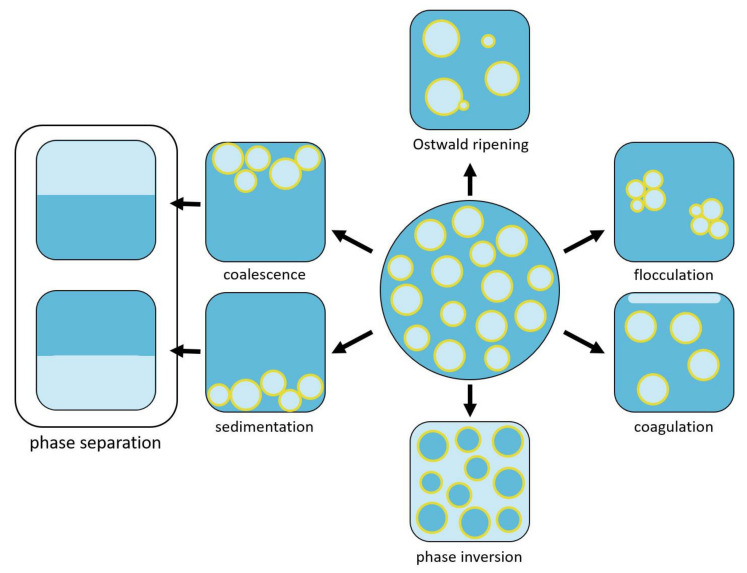
Representation of possible emulsion degradation pathways.

**Figure 3 polymers-14-01338-f003:**
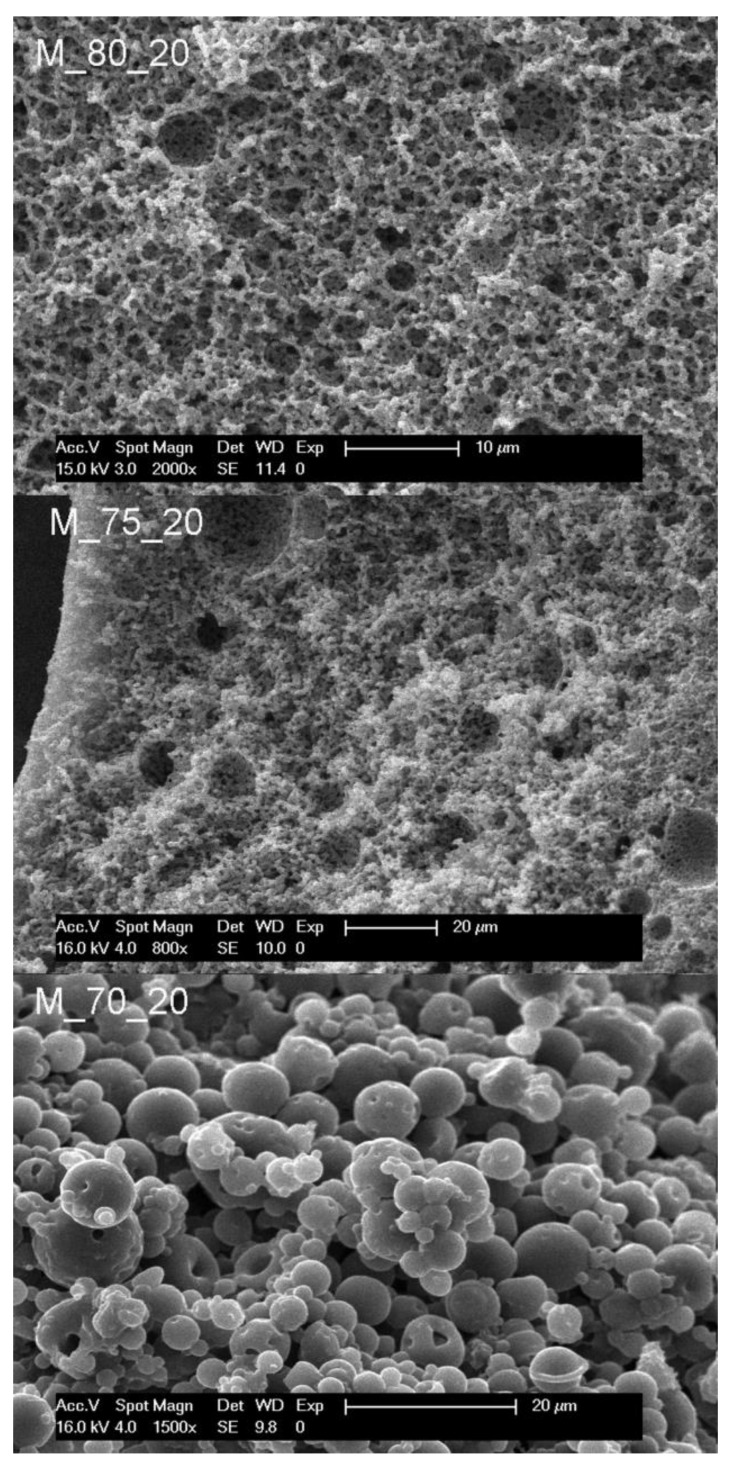
Scanning electron micrographs of samples prepared with 20% of surfactant.

**Figure 4 polymers-14-01338-f004:**
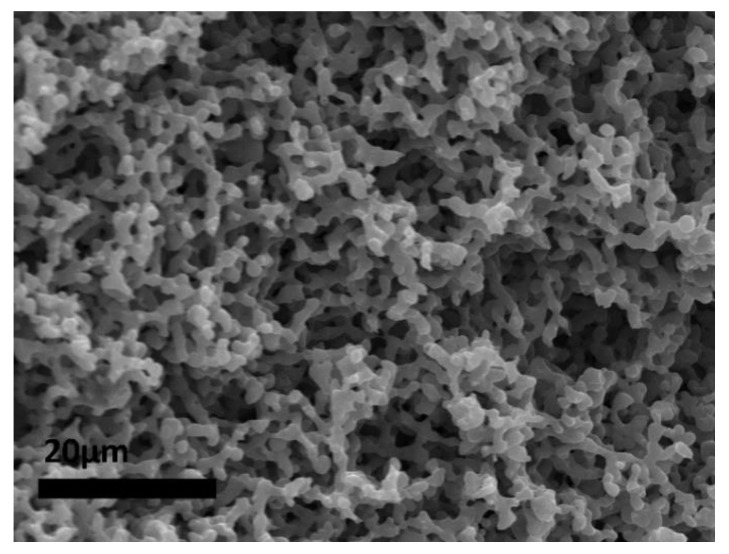
Scanning electron micrograph of sample M_75_20 (close up; bar is 20 µm).

**Figure 5 polymers-14-01338-f005:**
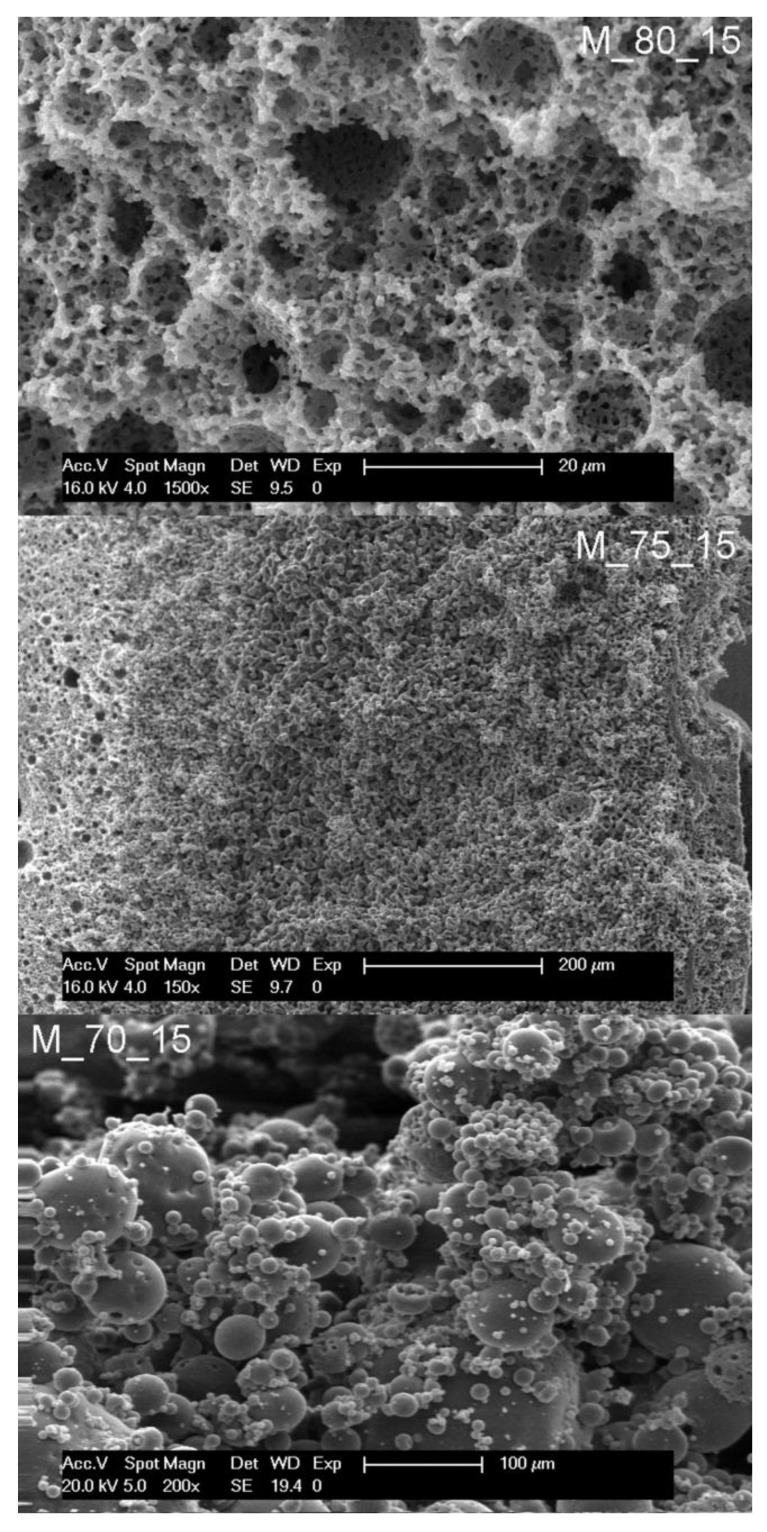
Scanning electron micrographs of samples prepared with 15% of surfactant.

**Figure 6 polymers-14-01338-f006:**
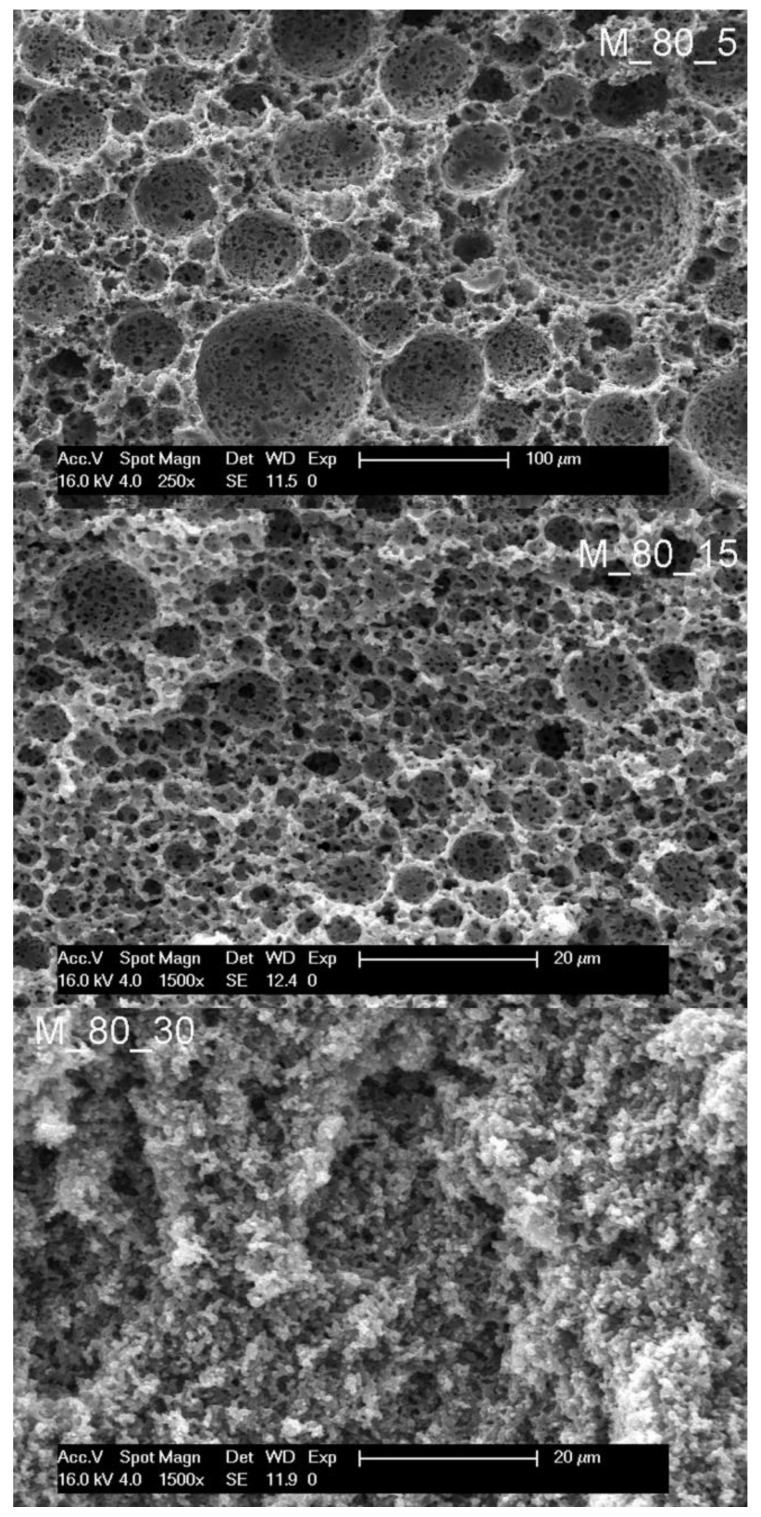
Scanning electron micrographs of samples prepared from emulsions with 80 vol% aqueous phase.

**Figure 7 polymers-14-01338-f007:**
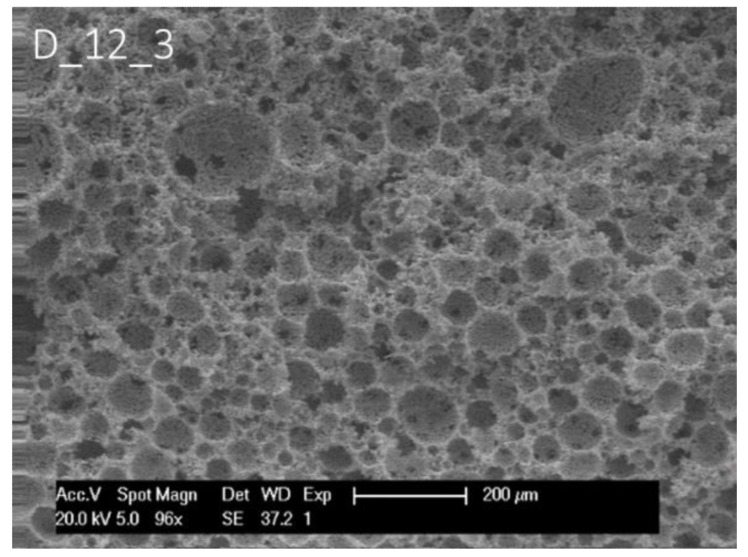
Scanning electron micrographs of samples prepared from DVA and TT with 15% surfactant.

**Figure 8 polymers-14-01338-f008:**
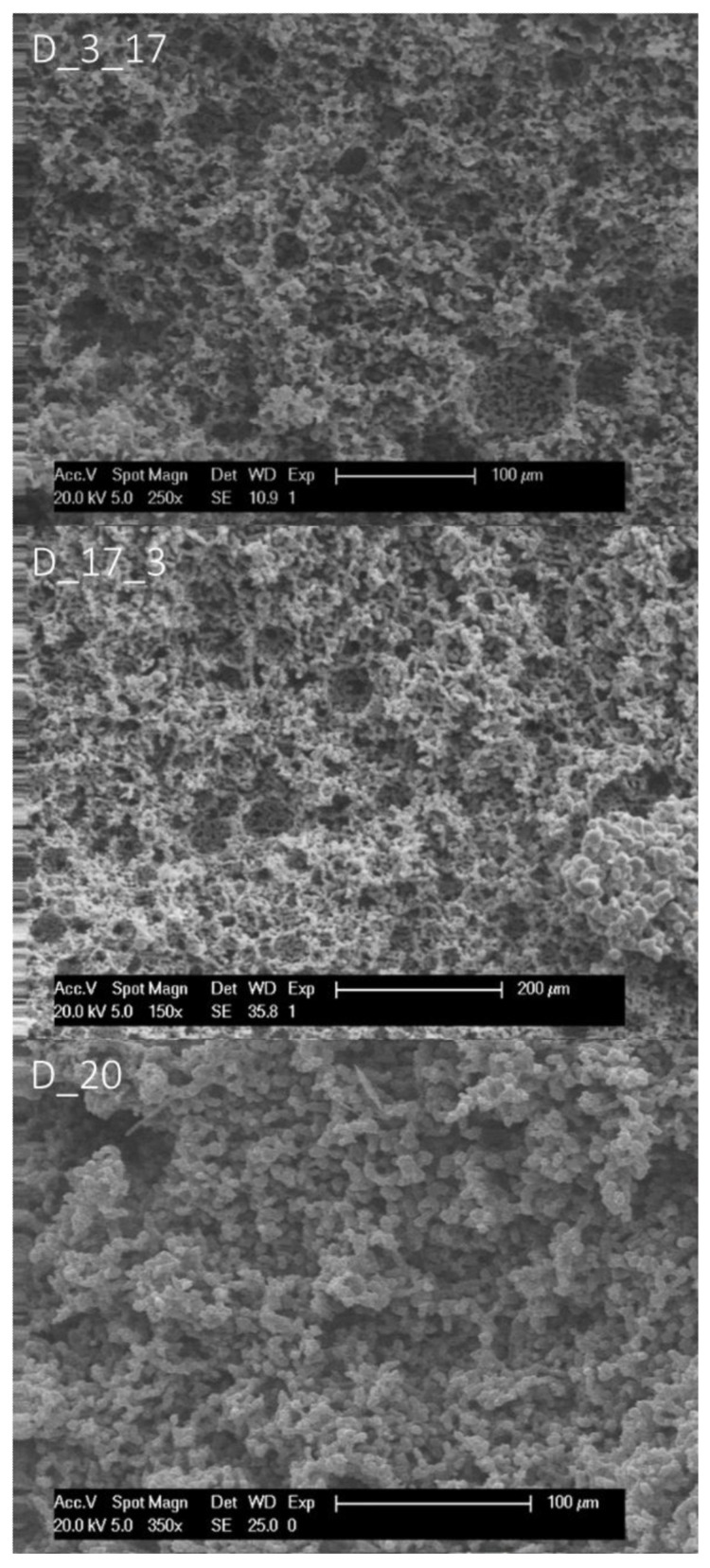
Scanning electron micrographs of samples prepared from DVA and TT with different surfactant compositions.

**Table 1 polymers-14-01338-t001:** Chemical composition of prepared HDDA/TT samples.

Sample	HDDA [g]	TT [g]	PEL-121 [%]	PEL-121 [g]	Internal Phase Volume [%]	Water Phase [mL]	Porosity * [%]
**M_80_20**	2.415	2.070	20	0.912	80	16	78
**M_75_20**	2.324	2.089	20	0.841	75	11.6	76
**M_70_20**	2.323	2.005	20	0.883	70	9	-
**M_80_15**	2.420	2.089	15	0.690	80	16	86
**M_75_15**	2.420	2.089	15	0.671	75	12	62
**M_70_15**	2.005	2.323	15	0.649	70	9	-
**M_80_5**	2.423	2.097	5	0.224	80	16	87
**M_80_15**	2.416	2.095	15	0.666	80	16	88
**M_80_30**	2.406	2.107	30	1.359	80	16	37

* Calculated porosity from pycnometry and envelope density methods.

**Table 2 polymers-14-01338-t002:** Chemical composition of prepared DVA/TT samples.

Sample	DVA [g]	TT [g]	Surfactant [%]	Surfactant Amount [g]	Internal Phase Volume [%]	Water Phase [mL]	Porosity * [%]
**D_12_3**	1.982	2.443	PEL-121 (12)	0.564	80	11.4	73
			Span 65 (3)	0.099			
**D_3_17**	1.982	2.443	PEL-121 (3)	0.133	80	11.4	74
			Span 65 (17)	0.752			
**D_17_3**	1.982	2.443	PEL-121 (17)	0.752	80	11.4	82
			Span 65 (3)	0.133			
**D_20**	1.982	2.443	PEL-121 (20)	0.885	75	10.7	66

* Calculated porosity from pycnometry and envelope density methods.

## Data Availability

Not applicable.

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
