# Peer review of "Emulsion Templated Porous Poly(thiol-enes): Influence of Photopolymerisation, Emulsion Composition, and Phase Behaviour on the Porous Structure and Morphology"

_polymers, 2022, doi:10.3390/polym14071338_

Round 1

Reviewer 1 Report

Dear Author, 

after completing my revision i could just say that there is almost no novelty in this work and it seem very similar to another work aldready published on Polymer Journal, as reported here. 

https://doi.org/10.1016/j.polymer.2017.04.021

This work in my opinion lacks of novelty, number of examples because it seems that the author just send this work whithout any other molecular example, and chemical charactherization.

thus, i reccomend to reject this work 

kindly

Author Response

After completing my revision i could just say that there is almost no novelty in this work and it seem very similar to another work aldready published on Polymer Journal, as reported here.

https://doi.org/10.1016/j.polymer.2017.04.021

This work in my opinion lacks of novelty, number of examples because it seems that the author just send this work whithout any other molecular example, and chemical charactherization.

thus, i reccomend to reject this work 

Response: Mentioned paper by Cameron et al. (Polymer 126, 395, 2017) is completely different to our report. Namely, the mentioned manuscript describes the preparation of polyHIPEs using thiol-ene and thiol-yne chemistry and determines the influence on mechanical properties and free thiol groups, studying the effect of thiol on the ratio of step growth vs. homopolymerisation. Our work also uses thiol-ene chemistry, however, the main line is the arising of bicontinuous and inverted morphology and discussing the mechanism behind it. The preparation of highly porous polymers with bicontinuous morphology from high internal phase emulsion has not been described before and the porous polymers with such morphology with features in the micrometre scale can have numerous applications, especially in the separation field when monolithic supports are used. This is highlighted throughout the text and especially in the conclusions: “We have shown that thiol/acrylate-based polymers synthesised via a thiol-ene radical-induced photochemical process within high internal phase emulsions can exhibit different porous morphologies, from a typical polyHIPE type cellular interconnected one, resulting from a kinetically stable high internal phase emulsion, through a bicontinuous with polymer/pores domains, to an inverted, hollow fused spheres like morphology. This is the first example of the preparation of a polymer monolith with bicontinuous morphology resulting from high internal phase emulsions. With careful emulsion composition setup, a thiol-ene photopolymerisation process can yield polymer monoliths with a bicontinuous morphology, high pore volume and homogeneous structure throughout the bulk of the monolith.”

We therefore cannot agree with the opinion that our paper lacks novelty and is a copy of the mentioned Cameron’s paper.

Reviewer 2 Report

Polymers

In their research article titled “Emulsion templated porous poly(thiol-acrylates): influence of  2 photopolymerization and emulsion composition on the porous structure and morphology” authors have investigated the 1,6-hexanediol  diacrylate  (HDDA)  and  pentaerythritol  tetrakis(3-mercaptopropionate)  (TT) were polymerized via a thiol-ene radical initiated photopolymerisation using emulsions with a high volume fraction of internal droplet phase and monomers in the continuous phase as precursors. The authors perform good quality work for the scientific community, I appreciate their efforts; however, modify your manuscript according to the following changes. I think it could be published with minor revisions.

Q 1 the caption of Figure 1 is not in a scientific way; please rewrite. And the same problem with figure 2.

Q 2 please also add some quantitative data in the abstract.

Q 3 The selection of references is too random. Authors should choose key references instead of a large stack of references.

Q 4 Write keywords properly

Q 5 All the chemicals used were bought from commercial sources" is this correct? Can we use commercial chemicals in laboratories? Answer this, please.

Q 6 figure 1 has a font difference to the text, and please amend it. Similar to figure 2, and so on.

Q 7 An emulsion can undergo a phase inversion process, apart from other destabilisation 

processes leading to phase separation (emulsion breakup; Figure 2). The phase inversion 

process  proceeds  through  a  meta-stable  bicontinuous-like  state. In  the  case  of  a 

microemulsion,  a  stable  bicontinuous  state  depends  on  the  surfactant  to  the  aqueous 

phase to oil phase volume ratio and on the surfactant structure. The bicontinuous state   can  be  used  for  templating  the  porosity  in  polymers  yielding  porous  polymers  with  a  bicontinuous topological structure. Numerous examples of the preparation of polymers  with bicontinuous morphology in this way are known; however, all in the domain of meso  and micropores due to the size dimensions of microemulsions. It is tough to understand this paragraph; please rewrite and add some latest references 10.1007/s42235-021-0046-7, 10.1149/1945-7111/ac0944, 10.1007/s10904-021-01942-1, and the following section or paragraph after that.

Author Response

In their research article titled “Emulsion templated porous poly(thiol-acrylates): influence of  2 photopolymerization and emulsion composition on the porous structure and morphology” authors have investigated the 1,6-hexanediol  diacrylate  (HDDA)  and  pentaerythritol  tetrakis(3-mercaptopropionate)  (TT) were polymerized via a thiol-ene radical initiated photopolymerisation using emulsions with a high volume fraction of internal droplet phase and monomers in the continuous phase as precursors. The authors perform good quality work for the scientific community, I appreciate their efforts; however, modify your manuscript according to the following changes. I think it could be published with minor revisions.

Q 1 the caption of Figure 1 is not in a scientific way; please rewrite. And the same problem with figure 2.

Response: The caption of Figure 1 was changed to »General scheme of porosity templating via high internal phase emulsion.” The caption of Figure 2 was changed to “Representation of possible emulsion degradation pathways”.

Q 2 please also add some quantitative data in the abstract.

Response: The data of porosity levels was added to the abstract.

Q 3 The selection of references is too random. Authors should choose key references instead of a large stack of references.

Response: There are 22 references in the manuscript; we believe this is not too many. Key references with regards to previous work in the field are noted.

Q 4 Write keywords properly

Response: Most characteristic keywords are chosen; we cannot see what is wrong with them.

Q 5 All the chemicals used were bought from commercial sources" is this correct? Can we use commercial chemicals in laboratories? Answer this, please.

Response: All chemicals are from commercial sources with defined purity. The producers are mentioned in the Experimental section.

Q 6 figure 1 has a font difference to the text, and please amend it. Similar to figure 2, and so on.

Response: Text within the figures is part of the graphic design, it is therefore normal not to be of the same font as the main text.

Q 7 An emulsion can undergo a phase inversion process, apart from other destabilisation 

processes leading to phase separation (emulsion breakup; Figure 2). The phase inversion 

process  proceeds  through  a  meta-stable  bicontinuous-like  state. In  the  case  of  a 

microemulsion,  a  stable  bicontinuous  state  depends  on  the  surfactant  to  the  aqueous 

phase to oil phase volume ratio and on the surfactant structure. The bicontinuous state   can  be  used  for  templating  the  porosity  in  polymers  yielding  porous  polymers  with  a  bicontinuous topological structure. Numerous examples of the preparation of polymers  with bicontinuous morphology in this way are known; however, all in the domain of meso  and micropores due to the size dimensions of microemulsions. It is tough to understand this paragraph; please rewrite and add some latest references 10.1007/s42235-021-0046-7, 10.1149/1945-7111/ac0944, 10.1007/s10904-021-01942-1, and the following section or paragraph after that.

Response: We believe that the paragraph is very clear- it tells what are the mechanisms behind the arousal of bicontinuous morphology and cites the most notable examples. Mentioned papers have no relevance or connection to the described process at all and therefore we don't see the possibility to include them in the reference list.

Reviewer 3 Report

Dear Editor and Authors,

I carefully read the paper, and unfortunately, the study presents a lack of originality, the analysis methods are not representative. In conclusion, I don’t recommend the publication of the paper in the present format.

Author Response

I carefully read the paper, and unfortunately, the study presents a lack of originality, the analysis methods are not representative. In conclusion, I don’t recommend the publication of the paper in the present format.

Response: As with the comment of reviewer 1, we highlite the novelty again.

The preparation of highly porous polymers with bicontinuous morphology from high internal phase emulsion has not been described before and the porous polymers with such morphology with features in the micrometer scale can have numerous applications, especially in the separation field when monolithic supports are used. This is highlited throughout the text and especially in the conclusions: “We have shown that thiol/acrylate-based polymers synthesised via a thiol-ene radical-induced photochemical process within high internal phase emulsions can exhibit different porous morphologies, from a typical polyHIPE type cellular interconnected one, resulting from a kinetically stable high internal phase emulsion, through a bicontinuous with polymer/pores domains, to an inverted, hollow fused spheres like morphology. This is the first example of the preparation of a polymer monolith with bicontinuous morphology resulting from high internal phase emulsions. With careful emulsion composition setup, a thiol-ene photopolymerisation process can yield polymer monoliths with a bicontinuous morphology, high pore volume and homogeneous structure throughout the bulk of the monolith.” Furthermore, the analysis methods are appropriate and representative- for studying the morphology, scanning electron microscopy, He picnometry and densiometry are most effective.

Round 2

Reviewer 1 Report

Dear Author,

again i am not convinced by this paper due to the fact that there is only one example. there is no mention about other reagents and substrates that demonstrate the quality of this work, if the author change a little the structure of the molecule the situation of their studies could change. one example is not enough!!!!

I recommend again to rejected this work in the present form

but if the authors could complete with other examples and with more and more exhaustive characterization it could be reconsidered. in this form it is not enough

Author Response

Dear Editor and reviewer,

as this is a communication paper it presents a major novelty in the form of bicontinuous morphology on the macroscopical level (several micrometers) from high internal phase emulsion precursors via the process of phase inversion. The characterization is done by scanning electron microscopy and a combination of skeletal density and overall density measurement to determine the exact porosity. We don't believe any further characterization would shed more light on the morphology of the prepared samples.

The reviewer suggests more chemicals and examples. We don't see how more chemicals and examples would improve the paper- as noted, this is a communication and as such focuses on one phenomenon rather than reporting a large number of experiments which would in this case bring no added value to the manuscript.

A number of factors affecting the resulting morphology has been presented- surfactant structure and concentration, temperature, phase volume ratio and morphologies resulting from the polymerisation of high internal phase emulsions clearly shown by SEM images.

Therefore we don't see how we can improve the manuscript by adding more examples.

Reviewer 3 Report

REject

Author Response

No comments by the reviewer supplied.

Round 3

Reviewer 1 Report

Dear Author,

I don't understand this focus of the authors, due to the fact that they could improve this work for example with other chemical reagents, such  as those reported in Polymer Chemistry by Cameron et al. 2014, 5, 6200. 

I think that the author could implement this work we other examples in order to see if their work and emulsion process can also work with other already existing polyHIPE.

I don't feel the completeness of the present work. 

Regards

Author Response

I don't understand this focus of the authors, due to the fact that they could improve this work for example with other chemical reagents, such as those reported in Polymer Chemistry by Cameron et al. 2014, 5, 6200. 

I think that the author could implement this work we other examples in order to see if their work and emulsion process can also work with other already existing polyHIPE.

I don't feel the completeness of the present work. 

Response:

In order to expand the choice of monomers, another system including a divinyl ether (divinyl adipate) was added to the description. It is shown that the behaviour of phase inversion resulting in a bicontinuous porous structure, is not limited to acrylate/thiol monomer mixture but behaves similarly in a vinyl ester/thiol mixture. Scanning electron microscopy images of resulting morphology (Figure 7, Figure 8) were added to the manuscript and Table 2, showing the composition of divinyl adipate based samples was also added. In the Results and Discussion section, description and discussion was added: »To study the effect of surfactant and surfactant composition on the morphological outcome structure of thiol-ene photopolymerized HIPEs, a different monomer composition was test-ed as well. The established system based on the vinyl ester, DVA and TT was first reported with two surfactants present in the HIPE formulation.[23] One way of achieving a macroporous bicontinuous structure would be to vary the ratio of surfactants or leave out one surfactant altogether to change in order to change the HLB and overall composition of the surfactant layer. It was chosen to lower the amount of Span 65 consecutively and final-ly omit it to have only PEL-121 present in the formulation. At first, the original setting with 15% surfactant was reproduced, which yielded a typical polyHIPE morphology, according to SEM (see Figure 7). SEM imaging revealed that the presence of Span 65 seems to be es-sential for establishing a cellular interconnected polyHIPE architecture (see Figure 8). When only PEL-121 was present, it was not possible to obtain a stable emulsion above 75 vol% of dispersed phase. However, with 75 vol% of internal phase, a monolith with bicon-tinuous structuremonolith was the result.

The comparison of the two systems, HDDA/TT and DVA/TT, highlights the importance of surfactant choice and composition in a HIPE. It also showed that upon careful tuning, it is possible to obtain a macroporous bicontinuous pore architecture from various thiol-ene HIPE formulations, opening a wider range of materials and material properties for poten-tial applications.«

Furthermore, to highlight the focus of the work, the termin »phase behaviour« was added to the title.

We hope you will find the manuscript satisfactorily ammended and thus fit for publication.

Round 4

Reviewer 1 Report

Dear Editor,

I appreciate the effort of author to enlarge the applicability of their system

Regards